# Larazotide acetate induces recovery of ischemia-injured porcine jejunum via repair of tight junctions

Zachary M. Slifer[1], Liliana Hernandez[1], Tiffany A. Pridgen[1], Alexandra R. Carlson[1], Kristen M. Messenger[1], Jay Madan[2], B. Radha Krishnan[2], Sandeep Laumas[3], Anthony T. Blikslager[1]*

1 Comparative Medicine Institute, Department of Clinical Sciences, North Carolina State University, Raleigh, NC, United States of America, 2 Innovate Biopharmaceuticals, Inc., Raleigh, NC, United States of America, 3 9 Meters Biopharma, Inc., Raleigh, NC, United States of America

* Anthony_Blkslager@ncsu.edu

**Data Availability Statement:** All relevant data are within the manuscript and its Supporting Information files.

## Abstract

Intestinal ischemia results in mucosal injury, including paracellular barrier loss due to disruption of tight junctions. Larazotide acetate (LA), a small peptide studied in Phase III clinical trials for treatment of celiac disease, regulates tight junctions (TJs). We hypothesized that LA would dose-dependently hasten recovery of intestinal ischemic injury via modulation of TJs. Ischemia-injured tissue from 6-8-week-old pigs was recovered in Ussing chambers for 240-minutes in the presence of LA. LA (1 μM but not 0.1 μM or 10 μM) significantly enhanced transepithelial electrical resistance (TER) above ischemic injured controls and significantly reduced serosal-to-mucosal flux LPS (*P<0.05*). LA (1 μM) enhanced localization of the sealing tight junction protein claudin-4 in repairing epithelium. To assess for the possibility of fragmentation of LA, an *in vitro* enzyme degradation assay using the brush border enzyme aminopeptidase M, revealed generation of peptide fragments. Western blot analysis of total protein isolated from uninjured and ischemia-injured porcine intestine showed aminopeptidase M enzyme presence in both tissue types, and mass spectrometry analysis of samples collected during *ex vivo* analysis confirmed formation of LA fragments. Treatment of tissues with LA fragments had no effect alone, but treatment with a fragment missing both amino-terminus glycines inhibited barrier recovery stimulated by 1 μM LA. To reduce potential LA inhibition by fragments, a D-amino acid analog of larazotide Analog #6, resulted in a significant recovery response at a 10-fold lower dose (0.1 μM) similar in magnitude to that of 1 μM LA. We conclude that LA stimulates repair of ischemic-injured epithelium at the level of the tight junctions, at an optimal dose of 1 μM LA. Higher doses were less effective because of inhibition by LA fragments, which could be subverted by chirally-modifying the molecule, or microdosing LA.

**Funding:** ATB, ZMS, Funded by Innovate Biopharmaceuticals Inc. (now 9 Meters Biopharma Inc.) https://9meters.com The funders were involved in study design and advice on revision of the manuscript. All final decisions were made by the authors at NC State University.

**Competing interests:** Anthony Blikslager and Zachary Slifer were funded, in part, by Innovate Pharmaceuticals to perform this work. Anthony Blikslager consulted for Innovate Biopharmaceuticals Inc and 9 Meters Biopharma. This does not alter our adherence to PLOS ONE policies on sharing data and materials.

**Abbreviations:** LA, Larazotide acetate; LPS, Lipopolysaccharide; F1, LA Fragment #1; F2, LA Fragment #2; F3, LA Fragment #3; F4, LA Fragment #4; A6, Analog #6; A6F1, Analog #6 Fragment #1; A6F2, Analog #6 Fragment #2; A6F3, Analog #6 Fragment #3; A6F4, Analog #6 Fragment #4; SD, single dose; MD, microdose; SEM, Standard error of the mean; TER, Transepithelial electrical resistance; DI, deionized; SIF, simulated intestinal fluid; AM, aminopeptidase M; HPLC, high performance liquid chromatography; TFA, trifluoroacetic acid; MS, mass spectrometry; TJ, tight junction; CeD, celiac disease; iFABP, intestinal fatty acid binding protein; LC-MS/MS, liquid chromatography/tandem mass spectrometric; FITC, fluorescein isothiocyanate; FITC-LPS, fluorescein isothiocyanate-labeled lipopolysaccharide; SDS-PAGE, sodium dodecyl sulfate-polyacrylamide gel electrophoresis; SEM, standard error of the mean; CP, carboxypeptidase P.

## Introduction

Intestinal epithelium exists as a monolayer that forms both transcellular and paracellular barriers to separate luminal contents from the subsurface interstitium. Apical tight junction (TJ) proteins are the primary regulators of paracellular barrier function, and their loss or dysregulation results in impaired barrier function. Celiac disease (CeD), a disease triggered by the ingestion and subsequent digestion of gluten with a worldwide prevalence of about 1% [1], includes pathophysiology consisting of compromised TJ proteins, which may result in the loss of paracellular barrier function. This loss of barrier function can lead to the exposure of luminal gliadin and antigenic interaction with immune cells within the lamina propria, furthering the dysregulation of TJ proteins and paracellular barrier dysfunction [2, 3]. Paracellular barrier loss can also occur after intestinal ischemic injury, which induces the loss of paracellular barrier function via disruption of TJs. This disruption of TJs following injury can result in the onset of sepsis and death in the host [4].

Larazotide acetate (LA) is an eight-chain amino acid investigational peptide (H-Gly-Gly-Val-Leu-Val-Gln-Pro-Gly-OH) and intestinal tight-junction regulator drug currently being studied in Phase III clinical trials in adult patients with CeD. LA was discovered through functional screening of the *Vibrio cholerae* enterotoxin, zonula occludens toxin (Zot), which is known to increase intestinal permeability through dysregulation of the intestinal tight junction structure. LA itself is derived from human zonulin, another known modulator of tight junction permeability that has structural similarities to Zot [5–7]. In Phase II clinical trials, LA limited the gluten-induced worsening of gastrointestinal symptom severity in patients assigned to LA versus those assigned to the study placebo [8, 9].

In addition to its efficacy in clinical trials, LA has been shown to have promising effects on protection of barrier function by regulating TJ proteins in other scientific studies [10–13]. Due to its action on TJs, we postulated that LA would also augment recovery of the TJ barrier after intestinal ischemic injury. However, scientific understandings of LA's actions during recovery of intestinal ischemia are unknown. Furthermore, its dose response and effect on mechanisms of acute repair after injury, including villous contraction, epithelial restitution, and TJ closure, have not been fully explored [14]. In this study, we investigated the effects of LA on mucosal recovery after segmental ischemic intestinal injury to determine if LA has applicability to this form of injury characterized by increased epithelial permeability.

## Methods

### Reagents

LA, A6 and their fragments were obtained from Innovate Biopharmaceuticals, Inc. (Raleigh, NC). LPS from *Escherichia coli* serotype 0111: B4 labeled with FITC (FITC-LPS) was purchased from Sigma.

### Animals

Six-to-eight-week-old Yorkshire crossbred pigs of either sex were obtained from the North Carolina State University Swine Educational Unit (Raleigh, NC). Over an approximate four-year period, 35 pigs were used across all experiments in this manuscript.

### Experimental porcine surgeries

All studies were approved by the North Carolina State University Institutional Animal Care and Use Committee. Six-to-eight-week-old Yorkshire crossbred pigs of either sex were individually housed and maintained on a commercially pelleted feed. Pigs were held off feed for 12

hours prior to experimental surgery. General anesthesia was induced with xylazine (1.5 mg/kg IM), ketamine (11 mg/kg IM), followed by mask delivery of isoflurane vaporized in 100% $O_2$. Pigs were subsequently orotracheally intubated, followed by continued delivery of isoflurane in 100% $O_2$ to maintain a surgical plane of anesthesia. An intravenous catheter was placed in an ear vein for delivery of fluids during surgery (lactated Ringer solution, 15 ml·kg$^{-1}$·h$^{-1}$, IV). The distal jejunum was approached via a ventral midline incision. Jejunal segments were delineated by ligating the intestine at 8–10 cm intervals with 2–0 silk and were subjected to segmental ischemia by ligating the local mesenteric blood supply with 2–0 silk for 45-minutes. Following the 45-minute ischemic period, pigs were euthanized with an intravenous overdose of pentobarbital prior to collecting the ischemic intestinal loops and uninjured control tissue.

## Ussing chamber studies

After 45-minutes of ischemia, the pigs were euthanized, and intestinal segments were collected. Non-ischemic control tissue was also collected at this time. Following tissue collection, the seromuscular layer was stripped from the mucosa while the tissue was bathed in oxygenated (95% $O_2$, 5% $CO_2$) Ringer's solution (in mmol/L: 154 $Na^+$, 6.3 $K^+$, 137 $Cl^-$, and 24 $HCO_3^-$; pH 7.4). The mucosa was subsequently mounted in 1.1-cm$^2$ aperture Ussing chambers, as described in previous studies [15, 16]. Tissues were bathed on both the mucosal and serosal sides with 10mL Ringer's solution. The mucosal bathing solution contained 10mM mannitol while the serosal side was osmotically balanced with 10mM glucose. Bathing solutions were oxygenated (95% $O_2$, 5% $CO_2$) and circulated in water-jacketed reservoirs. The spontaneous potential difference (PD) was measured using Ringer-agar bridges connected to calomel electrodes, and the PD was short-circuited through Ag-AgCl electrodes using a voltage clamp that was corrected for fluid resistance. TER ($\Omega$·cm$^2$) was calculated from the spontaneous PD and short-circuit current ($I_{sc}$). If the spontaneous PD was between -1.0 and 1.0 mV, tissues were current clamped at ± 100 μA for 5-seconds, and the PD was recorded. $I_{sc}$ and PD were recorded at 15-minute intervals over a 4-hour experiment. After tissues were mounted on Ussing chambers, all tissues were allowed to acclimate for a period of 30-minutes to establish stable baseline measurements, after which experimental, single-dose treatments of LA, LA fragments (F1, F2, or F3) with or without LA, A6, or A6 with or without A6 fragments (A6F1 or A6F2) were added to the apical chamber. For the microdosing regimen of LA, multiple doses at various concentrations of LA were added initially after the 30-minute acclimation period and every 45-minutes following this initial dose. Tissues were monitored by measuring transepithelial resistance (TER) for 240 minutes. Samples of Ringers solution were collected at select timepoints and quenched with 5% trifluoroacetic acid (TFA) solution (in 80% acetonitrile (ACN): 20% DI water). Quenched samples were centrifuged at 13000xg for 5-minutes and the resulting supernatants were stored at -80˚C for future MS analysis. Standardization of ischemic injury severity between intestinal loops was standardized using a similar method as previously described in another intestinal mucosal injury model [17]. First, T = 0 TER values from separate studies utilizing juvenile porcine intestinal ischemic tissue across three years were averaged (24 $\Omega$·cm$^2$), and the standard deviation of this dataset was calculated (11 $\Omega$·cm$^2$). Tissues with a T = 0 TER value exceeding one standard deviation of the mean (i.e., < 13 or > 35 $\Omega$·cm$^2$) were excluded from experiments and data analysis.

## LPS flux studies

LPS flux studies in Ussing chambers were carried out as previously described [18]. Briefly, after a 30-minute equilibration period, mucosal-to-serosal LPS fluxes were performed by adding 83 μg of lipopolysaccharide from *Escherichia coli* serotype 0111:B4 labeled with fluorescein

isothiocyanate (FITC-LPS) to the mucosal bathing solutions of uninjured control tissue, untreated ischemia-injured mucosa, and LA-treated, ischemia-injured mucosa and subsequently monitoring its appearance in the serosal bathing solutions. Aliquots of 200μl were collected in triplicate from the serosal side after 15-minutes (T45, baseline) and after 1-, 2-, and 3-hours following baseline and were then assessed for fluorescence.

## Histological examination

Histological examination was carried out as previously described [19]. Tissues from unrecovered control and ischemia-injured intestinal loops as well as recovered untreated and treated ischemia-injured mucosa were collected in 10% neutral buffered formalin for histological evaluation. Tissues were sectioned (5μm) and stained with hematoxylin and eosin. For each tissue, four well-oriented villi were identified in each representative serial section. The following measurements were gathered as previously described [4]. Villous height and villous width were calculated using ImageJ software (National Institutes of Health, Bethesda, MD). First, the scale bar provided on each imaged tissue section was used to create a baseline for micrometer measurements. A free-form line was centered in the middle of each villus and drawn from the base-to-tip of each villus to obtain total villous height. Villous width was calculated by drawing a line from left-to-right at the widest part of each villus. Denuded villous height was calculated using the same methods used to calculate total villous height with the exception of the measuring line stopping at the height in which epithelial cells were no longer present due to detachment and subsequent sloughing. The surface area of the villus was calculated by using the formula for the surface area of a cylinder. The formula was modified by subtracting the area of the base of the villus and multiplying by a factor accounting for the variable position at which each villus was cross sectioned. The percentage of the villous surface area that remained denuded was calculated, and the percent of denuded villous surface area was used as an index of epithelial restitution.

## Immunofluorescence histology

Immunofluorescence was carried out as previously described [4]. PFA-fixed tissues were transferred to 10% sucrose followed by 30% sucrose in 1X phosphate buffered saline (PBS) for 24 hours each for cryopreservation. Tissues were then embedded in optimal cutting temperature (OCT) compound and sectioned (7μm) onto positively charged glass slides for immunostaining. Slides were washed 3 times in 1X PBS to rehydrate the tissue and remove OCT compound then placed in 1X PBS-0.3% Triton-100 solution for 20-minutes to permeabilize the tissues. Tissues were washed twice in 1X PBS and incubated in Dako blocking solution (code # S0909, Agilent Technologies, Santa Clara, CA, USA) for 1 hour at room temperature followed by an overnight incubation at 4˚C with mouse anti-claudin-4 primary antibody (catalog # 32–9400, ThermoFisher Scientific, Waltham, MA) in Dako antibody diluent (code # S0809, Agilent Technologies, Santa Clara, CA) at a 1:500 working dilution. Following primary antibody incubation, tissues were washed 3 times with 1X PBS, and a goat anti-mouse secondary antibody (Alexa Fluor 568, catalog #A-11004, Invitrogen, Carlsbad, CA) diluted to a working dilution of 1:500 in Dako antibody diluent was applied to tissues for 1 hour at room temperature. Following 3 washes with 1X PBS, tissues were counterstained with nuclear stain 4',6-Diamidino-2-Phenylindol (DAPI, catalog # D3571, Invitrogen, Carlsbad, CA) diluted 1:1000 in antibody diluent for 5 minutes at room temperature. Images were captured using an inverted fluorescence microscope (Olympus IX81, Tokyo, Japan) with a digital camera (ORCA-flash 4.0, Hamamatsu, Japan) using 10X objective lens with numerical aperture of 0.3 (LUC Plan FLN, Olympus, Tokyo, Japan). Specificity of primary antibodies and lack of non-specific secondary antibody binding were confirmed by secondary only negative controls.

## Gel electrophoresis and western blot analysis

Gel electrophoresis and western blot analyses were carried out as previously described with slight modifications [15]. Unrecovered non-ischemic control and ischemia-injured mucosa, as well as tissues treated or untreated and recovered on Ussing chambers, were snap frozen and stored at -80˚C for preservation until SDS-PAGE and western analysis. Tissue aliquots were thawed at 4˚C and added to chilled lysis buffer containing Halt Protease and Phosphatase Inhibitor Cocktail (ThermoFisher Scientific, Waltham, MA) at 4˚C. Prior to SDS-PAGE, tissues were homogenized and fractionated using a Subcellular Protein Fractionation Kit for Tissues (ThermoFisher Scientific, Waltham, MA) into cytoplasmic and membrane fractions. Protein analysis of extract aliquots was performed (Pierce™ BCA Protein Assay Kit, ThermoFisher Scientific, Waltham, MA). Tissue extracts (amounts equalized by protein concentration) were mixed with an equal volume of SDS-PAGE sample buffer and boiled for 5-minutes. Lysates and protein standards (Precision Plus Protein™ Dual Color Standards, catalog # 1610374, Bio-Rad, Hercules, CA) were loaded into 4–12% Criterion™ XT Bis-Tris Protein Gels (Bio-Rad, Hercules, CA), and electrophoresis was carried out according to standard protocols. Proteins were transferred to a polyvinylidene fluoride (PVDF) membrane by using a Trans-Blot Turbo™ Transfer System (Bio-Rad, Hercules, CA). Membranes were blocked at room temperature for 60-minutes in Tris-buffered saline plus 0.05% Tween 20 (TBS-T) and 5% dry powdered milk or 5% bovine serum albumin (BSA), depending on primary antibody requirements. Membranes were washed with TBS-T and then incubated overnight in mouse anti-claudin 4 primary antibody (1:500, catalog # 32–9400, ThermoFisher Scientific, Waltham, MA); After an additional wash, membranes were incubated with horseradish peroxidase-conjugated secondary antibody (1:500, anti-mouse IgG (H+L), HRP Conjugate, Promega W4021) and developed for visualization of protein by the addition of enhanced chemiluminescence reagent (Clarity™ Western ECL Substrate, Bio-Rad, Hercules, CA). Following visualization and imaging of blots, membranes were stripped using Restore™ PLUS Western Blot Stripping Buffer (catalog # 46430, ThermoFisher Scientific, Waltham, MA) and followed the same protocol to probe overnight for a loading control using a mouse anti-β-actin primary antibody (1:1000, catalog # ab8226, Abcam, Cambridge, UK). Following this primary antibody incubation, the aforementioned protocol was continued using the same secondary antibody and visualization methods.

For aminopeptidase M detection in non-ischemic and ischemic jejunum, total protein concentrates were isolated from snap-frozen tissues using Total Protein Extraction Reagent (catalog # 78510, ThermoFisher Scientific, Waltham, MA). For a positive control of aminopeptidase M, porcine kidney aminopeptidase M (catalog # 164598, Sigma-Aldrich, St. Louis, MO) was used. Gel electrophoresis and western blot analysis were carried out as previously described in this manuscript. For primary antibody overnight-incubation, rabbit anti-CD13 primary antibody (catalog # a108310, Abcam, Cambridge, UK) was used, followed by secondary incubation with horseradish peroxidase-conjugated secondary antibody (goat anti-rabbit, catalog # 32460, Invitrogen, Carlsbad, CA) after necessary washes. Blot visualization and imaging was carried out using the aforementioned methods in this manuscript.

## *In vitro* enzyme degradation assay

Aminopeptidase M (AM) degradation of LA *in vitro* was carried out as follows. LA solution (0.2 mg/mL) in simulated intestinal fluid (SIF) (without Pancreatin), USP XXII Formulation: Ricca Chemical) pH 6.5 containing $MgCl_2$ (10 μM; catalyst support) was combined with AM (porcine kidney: Sigma) solution (final AM concentration is 0.66 unit/mL) in a vial. The combined mixture was incubated at 37˚C in a shaker water bath, and samples for UPLC-MS

analysis were collected at various timepoints quenched with 5% TFA solution (in 80% ACN: 20% DI water). Quenched samples were centrifuged at 13000xg for 5-minutes and the resulting supernatants were stored at -80˚C for future MS analysis.

## LC-mass spectrometry analytical method

A simple and sensitive liquid chromatography/tandem mass spectrometric (LC-MS/MS) method was developed for the identification and quantification of LA and its digestive fragments in the enzyme digest solution and other matrices. Samples were prepared with simple ACN and TFA in water quenching of the enzyme activity and precipitation of protein(s). The supernatant was then diluted appropriately and analyzed directly using the UPLC-MS/MS method Chromatographic separations were achieved on a C18 reversed phase column using an ACN and TFA-water gradient. The detection was performed in MS/MS mode via positive electrospray ionization (ESI+) interface. The method had a limit of detection (LOD) of 0.0125 μg/mL, a linearity range of 0.0125 to 5 μg/mL, precision of less than 12%, and accuracy within an acceptable range (±15%). This method was also demonstrated to be suitable for the analysis of LA in porcine intestinal and Ussing chamber fluids.

## LC-MS system

A Waters Quattro Xevo TOD mass spectrometer equipped with an electrospray ionization source and a Waters Acquity™ UPLC BEH system (Waters, Milford, MA, USA) with a binary solvent and a sample manager was used in this study. The LC column (Waters, C18, 50×2.1 mm, 1.7 μm) was operated at 35˚C and the autosampler was operated at ambient temperature. A binary mobile phase (mobile phase A and mobile phase B) with a linear solvent gradient system was used for the chromatographic separations of larazotide and its digestive fragments. Mobile phase A, 0.02% TFA in water and mobile phase B, 0.02% TFA in ACN. The UPLC-MS/MS method was qualified for the determination and identification of larazotide and the larazotide digestive fragments in the enzyme digest solution.

## Statistical analysis

Data were reported as means ± SEM. Data were analyzed by using either a one-way ANOVA or a two-way ANOVA for repeated measures for the effects of time and treatment (GraphPad Prism 8; GraphPad Software, San Diego, CA). A Tukey's test was used to determine differences between treatments following ANOVA. The threshold for statistical significance was set to 0.05.

# Results

## LA significantly enhances recovery of barrier function during recovery of ischemia-injured jejunal mucosa in a dose-dependent fashion

To determine if LA has an effect on barrier function during *ex vivo* recovery after intestinal ischemia, single doses of LA (0.1 μM, 1 μM, and 10 μM) were applied to the mucosal side of Ussing chambers after the equilibration period and were monitored for electrophysiological parameters (Fig 1A). Ischemic injury resulted in marked reductions in TER compared with uninjured control tissue at T = 0 (Fig 1A, $54.2 \pm 6.4 \ \Omega \cdot cm^2$ in control tissue vs. $25.6 \pm 5.8 \ \Omega \cdot cm^2$ in ischemia-injured jejunal mucosa, $P < 0.0001$). Application of 1μM LA to the mucosal surface of the ischemia-injured jejunum stimulated significant increases in TER at select timepoints (T = 180, T = 195, T = 240) throughout the recovery period when compared to untreated ischemia-injured mucosa (Fig 1A, $P < 0.05$ for all timepoints). Application of a

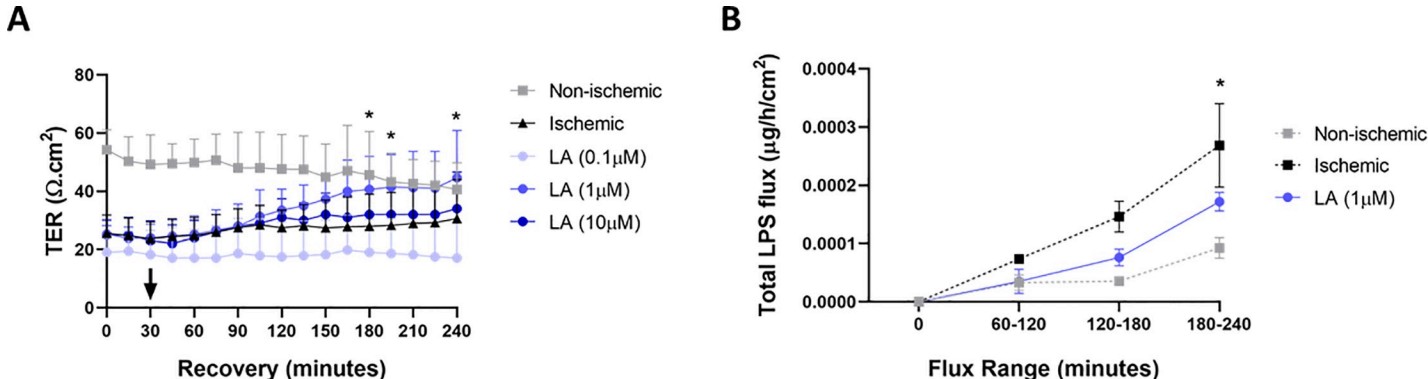

**Fig 1. Electrophysiological responses and FITC-LPS flux quantification of ischemia-injured porcine jejunal mucosa to treatment with single doses of LA at various concentrations.** LA was added to the apical bathing solution after a 30-minute equilibration period to allow for stabilization of baseline electrical measurements (indicated by the arrow at 30-minutes). Statistical significance (* or **) is shown between ischemia-injured jejunal mucosa treated with 1 μM LA and untreated ischemia-injured jejunal mucosa. TER Values are means ± SEM; n ≥ 7 except for LA (0.1 μM) where n = 3. Total LPS flux values are means ± SEM; n ≥ 9.

lower dose of LA (0.1 μM) did not stimulate significant recovery of ischemia-injured mucosa at any timepoint during recovery when compared to untreated ischemia-injured mucosa (Fig 1A, *NS*). Interestingly, a higher dose of LA (10 μM) also did not stimulate recovery of ischemia-injured mucosa during recovery (Fig 1A, *NS*). Mucosal-to-serosal flux of FITC-LPS during *ex vivo* recovery of ischemia-injured jejunum showed a similar pattern, with treatment of ischemia-injured mucosa with 1 μM LA significantly reducing FITC-LPS flux when compared to untreated ischemia-injured jejunal mucosa during the 4-hour recovery period (Fig 1B, T = 240, *P < 0.05*).

## LA (1 μM) does not affect villous contraction or epithelial restitution during *ex-vivo* recovery of ischemia-injured jejunal mucosa

To determine if LA had an effect on microscopically measurable parameters of repair (villous contraction and restitution), histomorphologic imaging and histomorphometric analyses were conducted. After initial ischemic injury, detachment of the epithelial monolayer from the basement membrane and subsequent sloughing of cells was observed as compared to adjacent uninjured tissue (Fig 2A, 2B). Following the 4-hour *ex vivo* recovery period, villous heights for ischemia-injured tissues treated with 1μM LA were no different from villous heights of untreated ischemia-injured jejunum (Table 1, 182.8 ± 8.1 μM in ischemia-injured jejunum treated with 1 μM LA vs. 175.3 ± 9.5 μM in untreated ischemia-injured jejunum, NS). Additionally, both untreated ischemia-injured (Fig 2C) and LA-treated (Fig 2D) mucosal tissues exhibited similar levels of epithelial restitution after the 240-minute recovery period, as denoted by a significant reduction in denuded villous surface area after the *ex vivo* recovery period (Table 1, 100 ± 0.0% epithelialization in 1μM LA treated ischemia-injured jejunum and 98.5 ± 1.3% in untreated ischemia-injured jejunum, *P < 0.0001* for both groups when compared to unrecovered ischemic control tissue at 77.8 ± 7.2%).

## LA (1 μM) has an effect on tight junction proteins

To initially determine if LA had an effect on TJ proteins during recovery of ischemia-injured intestine, immunofluorescence staining of LA-treated and untreated tissues recovered after ischemic injury was conducted using a claudin-4 primary antibody probe. Immunofluorescence imaging demonstrated qualitatively more visible claudin-4 in the membrane of

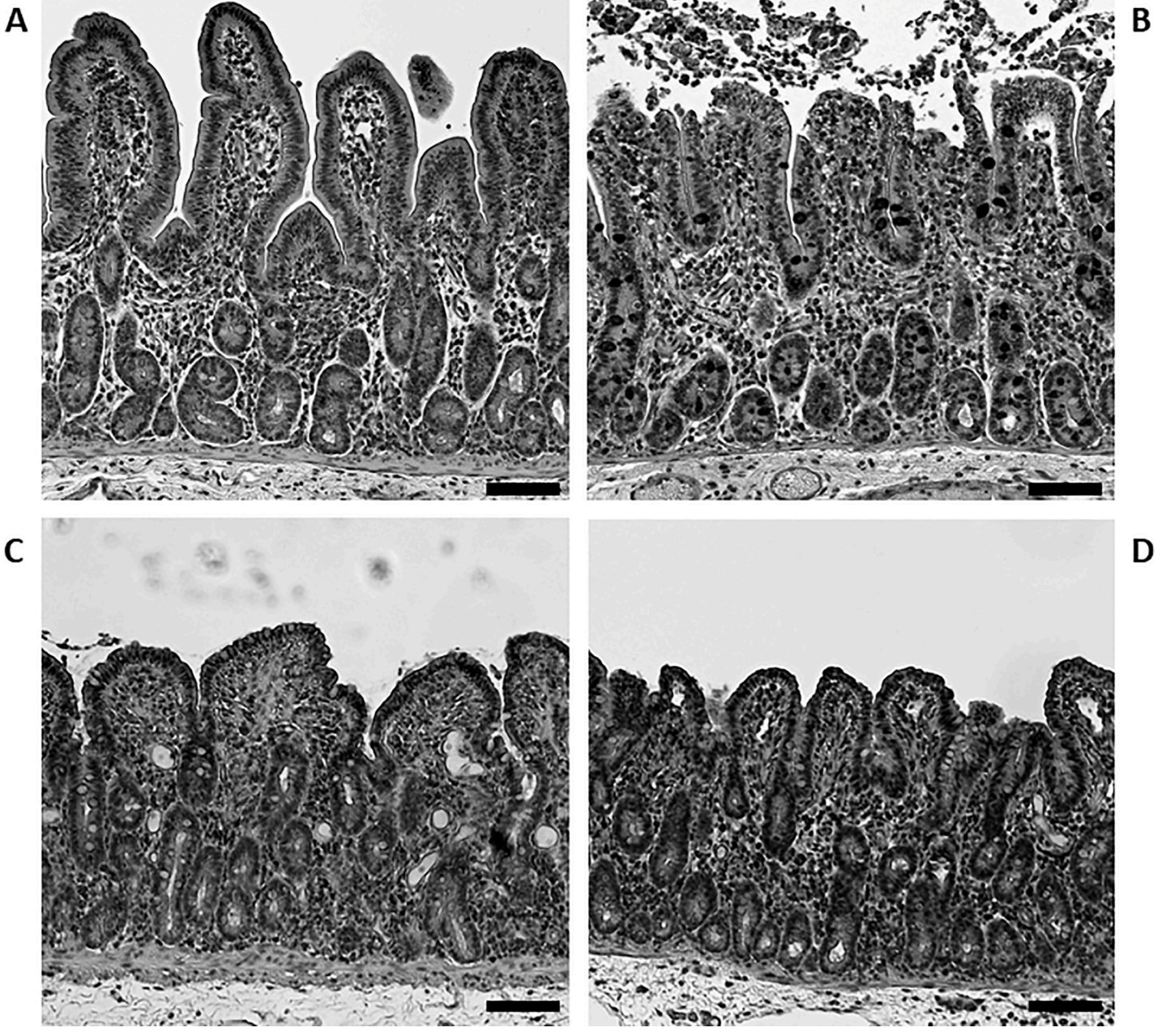

**Fig 2. Histological examination of porcine jejunal mucosa subjected to ischemia and *ex vivo* recovery.** A: Unrecovered non-ischemic (uninjured) porcine jejunum. B: unrecovered ischemia-injured porcine jejunum. C: Recovered ischemia-injured porcine jejunum. D: Recovered ischemia-injured porcine jejunum treated with LA (1 μM). Ischemic injury resulted in the detachment of the epithelial monolayer from the basement membrane and sloughing of epithelium into the intestinal lumen (2B). After 4 hours of recovery, there were no histological differences between ischemia-injured jejunum treated with 1 μM LA (2D) versus untreated ischemia-injured jejunum (2C). Villous height appeared to be similar between groups, and no denuded surface was observed in either group. Hematoxylin and eosin stain, 100 μM scale bar.

recovered intestine treated with LA when compared to untreated, ischemia-injured intestine after 240 minutes of recovery (Fig 3). Subsequent western blot analysis of cytoplasmic and membrane protein fractions from LA-treated ischemia-injured tissues revealed significantly increased claudin-4 protein in the membrane fraction when compared to untreated ischemia-injured tissues after 240 minutes of recovery (Fig 4B, Claudin-4/β-actin densitometry ratio of

**Table 1. Histomorphometric assessment of unrecovered and recovered ischemia-injured porcine jejunum.**

| Treatment | Villous Height (µM) | | Villous Width (µM) | | Epithelialization (%) | |
|---|---|---|---|---|---|---|
| | Unrecovered | Recovered | Unrecovered | Recovered | Unrecovered | Recovered |
| Non-isch | 290.7 ± 24.2 | 214.4 ± 22.7 | 130.7 ± 5.9 | 94.7 ± 13.3 | 100 ± 0.0 | 100 ± 0.0 |
| Isch | 195.6 ± 31.0 | 175.3 ± 9.5 | 106.5 ± 3.7 | 106.6 ± 11.8 | 77.8 ± 7.2 | 98.5 ± 1.3 |
| LA (1 µM) | N/A | 182.8 ± 8.1 | N/A | 116.7 ± 17.0 | N/A | 100.0 ± 0.0 |

Values are represented as averages plus or minus the SEM.

0.80 ± 0.05 in LA-treated, ischemia-injured tissues vs. 0.43 ± 0.06 in untreated, recovered ischemia-injured tissues, *P < 0.01*). There were no statistical differences in cytoplasmic claudin-4 when comparing LA-treated tissues to untreated ischemia-injured tissues after recovery (Fig 4B, Claudin-4/β-actin densitometry ratio of 0.75 ± 0.05 in LA-treated tissues vs. 0.81 ± 0.06 in untreated ischemia-injured tissues, *NS*).

## LA is degraded *in vitro* by the intestinal brush-border enzyme aminopeptidase M

To determine if LA can be fragmented by the brush-border enzyme aminopeptidase M (AM) [20], an *in vitro* enzyme assay was carried out in two separate experiments in which LA was incubated in the presence of AM. In both enzymatic experimental runs, LA was fragmented throughout the entirety of the experiment (Fig 5A and 5B). Additionally, F1 (N-deGlycine) and F2 (N-di deGlycine) were both generated throughout the experiment, and both fragments were found in all samples collected during the run, with the exception of F2 not being detected after 90 minutes. This suggests that the formed fragments F1 and F2 are further degraded to smaller fragments over time. Western blot analysis of unrecovered non-ischemic (uninjured) and ischemic (injured) porcine jejunum revealed the presence of AM in both tissues (Fig 5C), suggesting that fragments of LA during *ex vivo* recovery are likely generated, at least in part, by AM.

## Fragments of LA are generated during *ex vivo* recovery after intestinal ischemia

To determine if LA is fragmented by aminopeptidase M during *ex vivo* recovery, Ringers solution samples collected for MS analysis were analyzed. Mass spectrometry analysis of these samples revealed that both LA fragments F1 and F2 are formed during this *ex vivo* recovery process (Fig 6). In addition to F1 and F2, two other LA fragments were detected upon MS analysis, F3 (C-deGlycine) and F4 (all deGlycine) with F3 having the highest concentration out of all four fragments (Fig 6).

## LA fragment F2 but neither F1 nor F3 inhibits recovery of ischemia-injured jejunal mucosa

After *ex vivo* formation of LA fragments was revealed, an overdose, 10 fold excess, of F1 (10 µM), F2 (10 µM), or F3 (10 µM) with or without LA (1 µM) was added to the apical bathing solution of recovering ischemia-injured jejunum to determine if these fragments have an effect on *ex vivo* recovery. Application of 10µM F1 or F3 in the presence of 1µM LA did not significantly inhibit *ex vivo* recovery of ischemia-injured jejunal mucosa throughout the 4-hour recovery period when compared to injured mucosa treated with 1 µM LA alone (Fig 7A and 7C, *NS*). However, the recovery response of ischemia-injured jejunum in the presence of 1µM

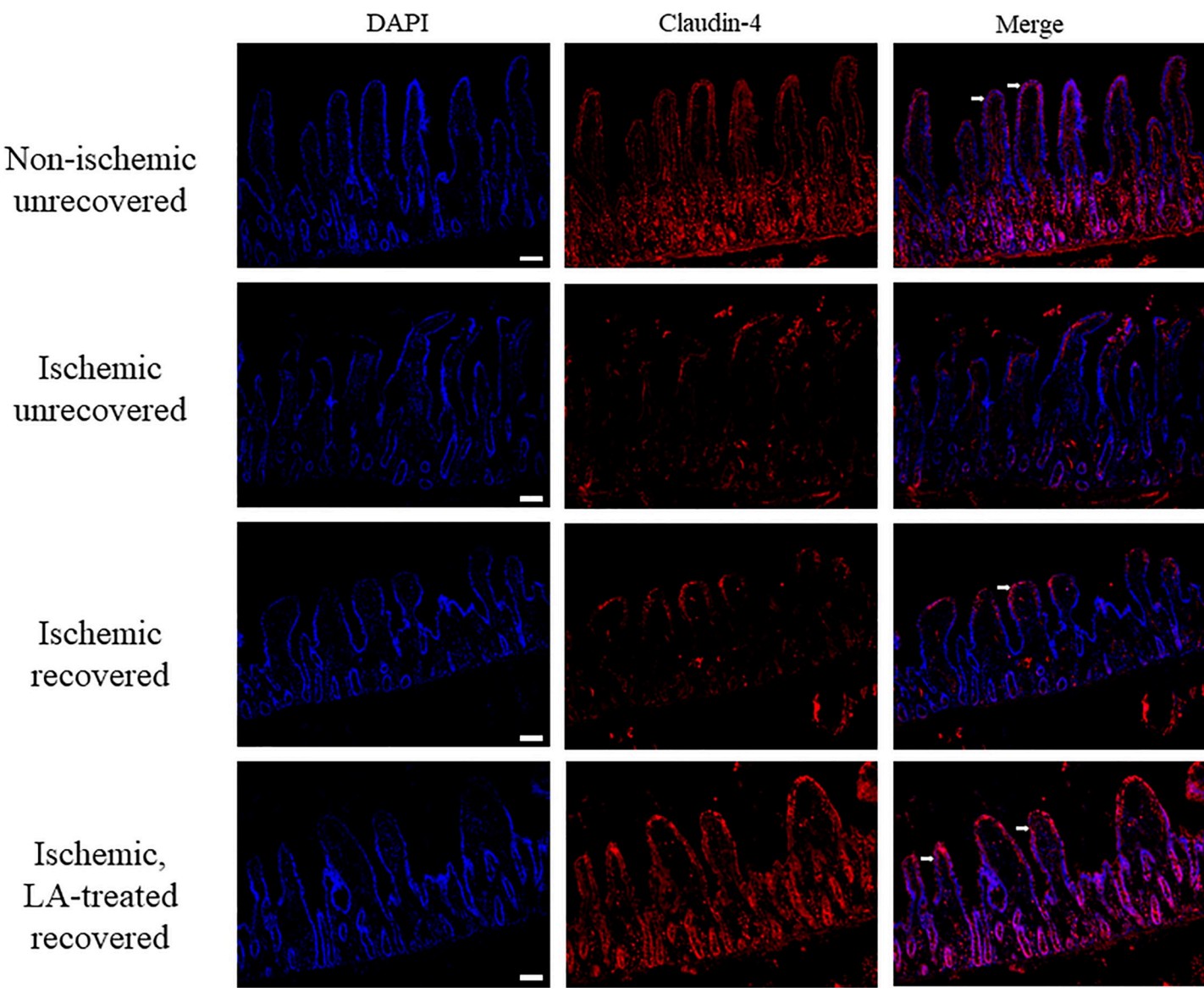

**Fig 3. Immunofluorescent analysis of claudin-4 of ischemia-injured porcine jejunum after LA (1 μM) treatment.** Arrows indicate where claudin-4 is well localized within the epithelium. Scale bar represents 100 μM.

LA was significantly inhibited by treatment with 10μM F2 at select timepoints (T = 195, T = 210, T = 240) throughout recovery when compared to tissues treated only with LA (Fig 7B, T = 195, *P < 0.05; T = 210, P < 0.05; T = 240, P < 0.01*). Treatment with either F1 alone or F2 alone exhibited significantly reduced TER when compared to treatment with 1 μM LA alone but were not significantly different from untreated, ischemia-injured tissues at any timepoint during recovery.

## Analog #6 (A6), a D-amino acid analog of larazotide, enhances recovery of barrier function during recovery of ischemia-injured jejunal mucosa

Because of identified complications of fragmentation of LA peptide, we explored the effect of chirally-modified LA molecule A6 with all D-amino acids on barrier function during *ex vivo*

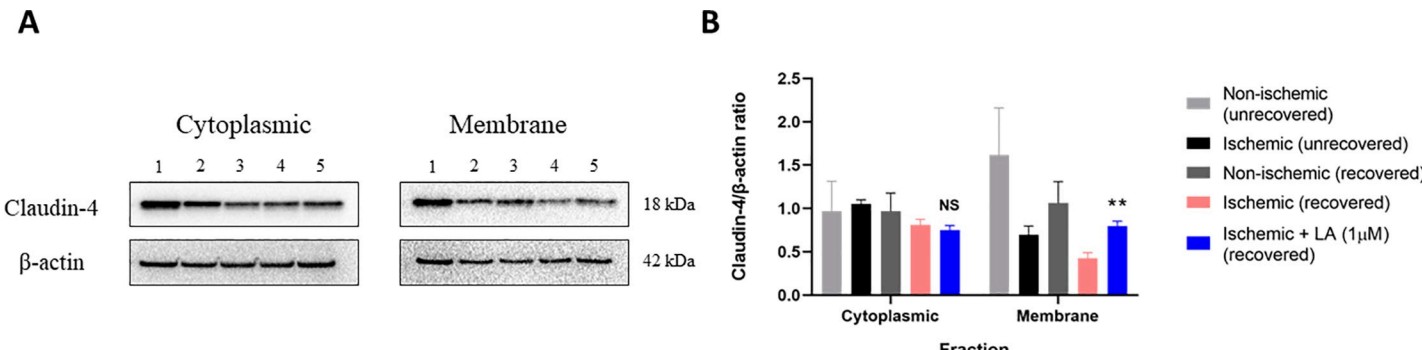

**Fig 4. Expression of tight junction proteins in LA-treated ischemia-injured porcine jejunum.** A: Representative western blot images from cytoplasmic and membrane fractions from ischemia-injured porcine jejunum. Lanes for panel A are as follows: 1: non-ischemic (unrecovered), 2: ischemic (unrecovered), 3: non-ischemic (recovered), 4: ischemic (recovered), 5: ischemic + 1 μM LA (recovered) B: Densitometry analysis of western blot bands for claudin-4 from cytoplasmic and membrane protein fractions of represented tissues. Claudin-4 expected band size using this monoclonal antibody is ~18 kDa. β-actin loading control band size is expected at ~42 kDa. Statistical significance (**) is between ischemic + LA (1 μM) (recovered) tissues vs untreated ischemic (recovered) tissues. Values are means ± SEM; n = 3 for each group.

**Fig 5. *In vitro* degradation of LA by AM.** Two separate runs (A & B) are shown. Each graph represents data from samples collected from LA, MgCl₂, and AM solution mixtures. Control data from samples not containing AM are not shown. C: Western blot analysis of uninjured and injured (unrecovered) porcine jejunum for AM expression.

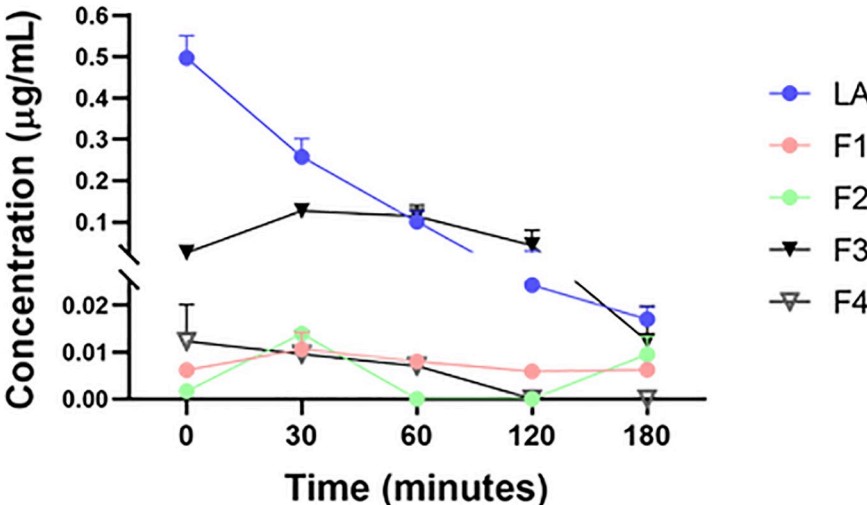

**Fig 6. Mass spectrometry analysis of Ringers solution collected from the Ussing chambers (N = 4) during *ex vivo* recovery in the presence of LA.** A: LA fragments F1 and F2 are generated during the 240-minute *ex vivo* recovery. Additionally, two more LA fragments, Fragment #3 (F3) and Fragment #4 (F4) were also detected upon MS analysis of the same Ringers solution samples.

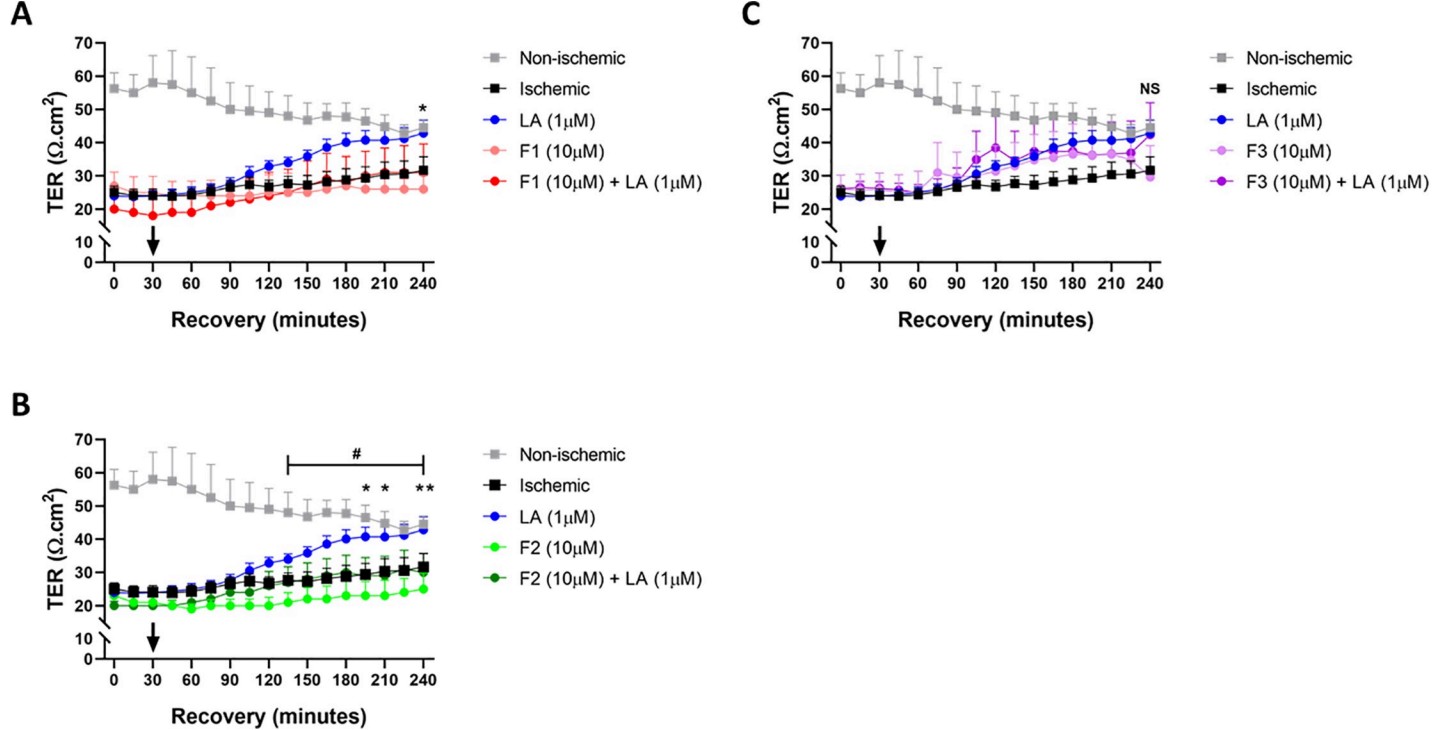

**Fig 7. Electrophysiological responses of ischemia-injured porcine jejunal mucosa to treatment with A: LA fragments F1 alone or with LA, B: F2 alone or with LA, and C: F3 alone or with LA.** Values are means ± SEM; n ≥ 3. F1/F2 alone or with LA was added to the apical bathing solution after a 30-minute equilibration period to allow for stabilization of baseline electrical measurements. A: Statistical significance (*) is shown between F1 (10 μM) alone vs LA (1 μM). B: Statistical significance (* or **) is shown between F2 (10 μM) + LA (1 μM) vs LA (1 μM). Additional statistical significance (#) is shown between F2 (10 μM) alone vs LA (1 μM). C: No statistical significance was detected between groups at specific timepoints. Values are means ± SEM; n ≥ 3.

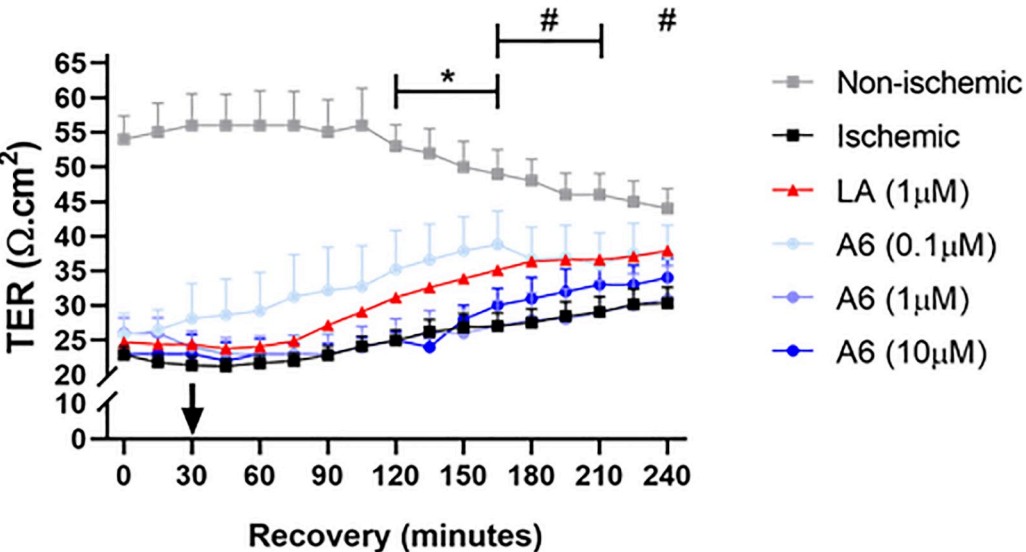

**Fig 8. Electrophysiological responses and histomorphology of ischemia-injured porcine jejunal mucosa to treatment with single doses of the chirally-altered LA molecule A6 at various concentrations.** Statistical significance is between either ischemic tissue treated with A6 vs untreated ischemic tissue (*) or ischemic tissue treated with LA vs untreated ischemic tissue (#) during the 240-minute recovery period. Values are means ± SEM; n ≥ 4.

recovery after intestinal ischemia. Single doses (0.1 μM, 1 μM, and 10 μM) of A6 were applied to the mucosal side of Ussing chambers after the equilibration period and were monitored for electrophysiological parameters (Fig 8). While the application of 1 μM and 10 μM of A6 on ischemia-injured jejunum did not result in significantly different TER recovery profiles when compared to ischemia-injured tissues treated with LA (1 μM), the TER changes throughout recovery were also not significantly different from untreated ischemia-injured tissues. Interestingly, the single dose of 0.1 μM of A6 added to the mucosal surface of the ischemia-injured jejunum stimulated significant increases in TER when compared to untreated ischemia-injured tissues at select timepoints during recovery (Fig 8, T = 120, T = 135, T = 150, T = 165, *P < 0.05* for all). Additionally, 0.1 μM of A6 exhibited these significant increases in TER earlier (starting at T120) than LA-treated ischemia-injured tissues (starting at T165) (Fig 8).

## Fragment A6F1 of chirally-modified LA (A6 fragments) inhibits recovery of ischemia-injured jejunal mucosa

Initially, Ringers solution samples were analyzed to verify whether or not A6 fragments were generated during recovery. Mass spectrometry analysis of these samples revealed the formation of three A6 fragments during the *ex vivo* recovery process: A6F1 (N-deGlycine), A6F2 (N-di deGlycine), and A6F4 (all deGlycine) where A6F1 was the predominant fragment formed (Fig 9). A6F3 was not detected in the measured samples. Following the detection of A6 fragments, single doses of the aforementioned fragments (10 μM), 10-fold excess, were applied to the mucosal surface of the ischemia-injured jejunum in the absence or presence of A6 (0.1 μM). Application of 10 μM A6F1 in the presence of 0.1 μM A6 inhibited *ex vivo* recovery throughout the recovery period when compared to treatment with 0.1 μM A6 alone (Fig 10A, T = 150, *P < 0.05*; T = 165, *P < 0.05*). Treatment of ischemic tissues with A6F2 (10 μM) in the presence of A6 (0.1 μM) during recovery resulted in a similar recovery to ischemic control tissue but was not significantly different from tissues treated with only 0.1 μM A6 (Fig 10B, 10 μM A6F2 with 0.1 μM A6 vs 0.1 μM A6 alone, *NS*). Interestingly, treatment with A6F2 (10 μM) alone

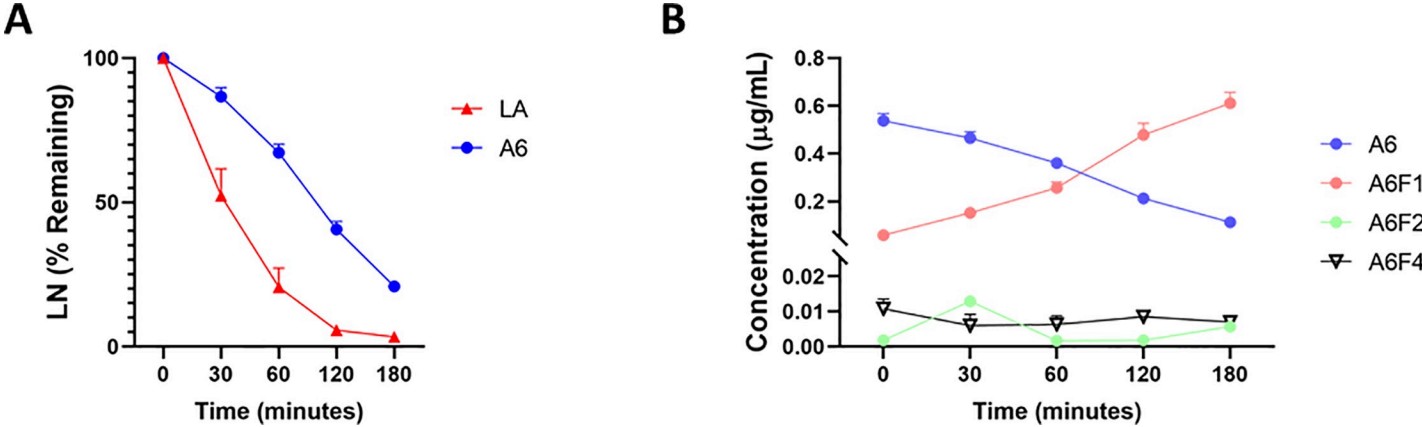

**Fig 9. Mass spectrometry analysis of Ringers solution collected from the Ussing chambers (N = 4) during *ex vivo* recovery in the presence of A6 or LA.** A: Degradation of either LA or A6 during *ex vivo* recovery. The half-life of LA and A6 in ischemic injured jejunum tissue is 35 minutes and 78 minutes, respectively. B: A6 fragments A6F1, A6F2, and A6F4 are generated during the 240-minute *ex vivo* recovery. Values are means ± SEM; n = 4.

resulted in similar increases in TER compared to 0.1 μM A6 alone, although this recovery profile of A6F2 (10 μM) was not significantly higher than the untreated, ischemia-injured tissue TER (Fig 10B, 10 μM A6F2 vs untreated, ischemia-injured tissue, *NS*).

As an additional means of potentially reducing the concentration of inhibitory LA fragments, preliminary experiments of a microdosing regimen of 0.1 μM LA every 45-minutes starting at T = 30 revealed a similar recovery profile when compared to a single dose of 1 μM LA (S1 Fig). However, further experiments with additional dosing regimens should be carried out to determine significant effects of LA microdosing.

## Discussion

In the present study, we demonstrated the restorative effect of LA on disrupted barrier function during *ex vivo* recovery after intestinal ischemic injury. Intestinal ischemic injury is typically characterized by detachment of the epithelium from the basement membrane, compromised interepithelial junctions, and cell sloughing [21, 22]. During acute mechanisms of repair after intestinal ischemia, villi contract to reduce the denuded surface area over which epithelial cells must restitute, and epithelial restitution occurs to resurface the denuded villous surface. Once restitution is complete, closure of interepithelial junctions via re-sealing of TJ

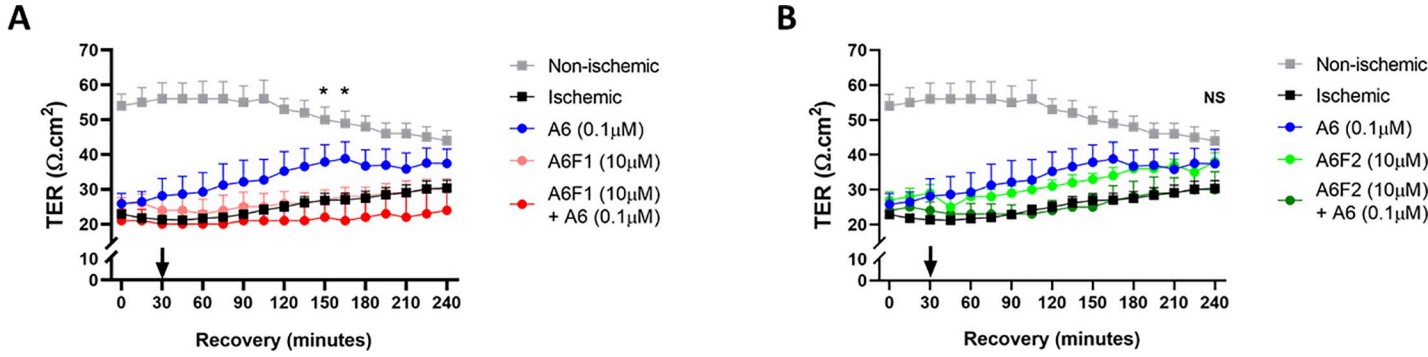

**Fig 10. Electrophysiological responses of ischemia-injured porcine jejunal mucosa to treatment with A: A6F1 alone or with A6 and B: A6F2 alone or with A6.** Statistical significance is between A6F1 (10 μM) + A6 (0.1 μM) vs A6 (0.1 μM) alone. Values are means ± SEM; n ≥ 4.

proteins is critical to restore mucosal barrier function. In this study, we determined that LA does not play a role in villous contraction or epithelial restitution, as indicated by similar villous height and re-epithelialization levels between recovered LA-treated and untreated injured intestine. However, restorative effects of LA were revealed upon analysis of electrophysiological and mucosal-to-serosal LPS flux data, specifically by showing significant elevations of TER and significant reductions in FITC-LPS flux levels in the presence of LA as compared to untreated ischemic-injured tissues. These data indicate an effect of LA on reduced paracellular permeability during recovery.

This effect of LA on the TJ-restoration step during acute mechanisms of repair after intestinal ischemia was further supported by IF and western blot findings of claudin-4 membrane expression being significantly elevated when compared to untreated, recovered ischemic intestine in the western blot analysis. This data indicates that LA modulates recycling of the sealing tight junction protein claudin-4 back to the tight junction complex in the membrane as at least a partial explanation of its mechanism of action. It is simultaneously important to recognize that the tight junction complex is not made up by a single tight junction protein, such as claudin-4, as it consists of a variety of protein elements including transmembrane claudins and occludin proteins, tight junction-associated transmembrane molecules, and intracellular scaffold proteins [23]. However, we believe claudin-4 may serve as a major regulatory tight junction protein during intestinal ischemic injury due to its suggested highest expression in villous surface epithelium of the small intestine combined with the known pathological progression of intestinal ischemia [24]. Previous literature describes intestinal ischemia as impacting the villous tips first due to the luminal surface epithelium experiencing physiological hypoxia, which stems from a decreasing gradient of oxygen moving from the crypt to the villus [25]. This physiological hypoxia towards the tips of villi increases susceptibility of villous epithelium to injuries such as intestinal ischemia, resulting in detachment of villous epithelium from the basement membrane, dissociation of interepithelial junctions, and subsequent sloughing of epithelium into the intestinal lumen [25]. While there are still more possibilities for tight junctional complex targets that may also be regulated by LA during recovery of ischemia-injured intestine, there is potential that restoration of barrier function after intestinal ischemia involves the regulation of claudin-4 back into the membrane of villous epithelium given the findings in this manuscript.

Although changes in membrane expression of claudin-4 indicate a role of LA in regulation of TJ proteins during recovery of ischemic-injured intestine, the mechanisms by which LA regulate TJs remain incompletely understood. One potential pathway in which LA may regulate tight junction structure is through the myosin light chain (MLC) pathway. Phosphorylation of MLC has been shown to open tight junctions via contraction of the perijunctional actomyosin ring that is linked to transmembrane tight junction proteins, resulting in increased permeability [16, 26]. In a cell culture model of anoxia/reoxygenation, monolayers of Caco-2 brush border-expressing (C2BBe1) cells that were pre-treated with LA prior to anoxia exhibited significantly reduced phosphorylated myosin light chain-2 (pMLC-2) and contained well-organized TJ proteins when compared to untreated C2BBe1 cells during reoxygenation [27]. These data suggest LA regulates MLC phosphorylation, thereby allowing for the closure of interepithelial TJ proteins.

An alternative mechanism by which LA regulates tight junctions is through affecting the endogenous zonulin pathway. Gliadin peptides can bind to chemokine CDC motif receptor 3 (CXCR3) upon reaching the lumen of the intestine, which will eventually result in the release of zonulin into the lumen [2]. Zonulin can then bind to its receptors, epidermal growth factor receptor (EGFR) and protease-activated receptor 2 (PAR2), which will cause TJ disassembly [2, 28, 29]. It is believed that LA works to antagonize the endogenous zonulin pathway through

preventing zonulin permeating activity, therefore preventing the disassembly of tight junctions [30]. However, this pathway has not been assessed to our knowledge in tissue recovery from acute injury such as ischemia.

An interesting finding during this study was that a higher dose of LA (10 μM) did not elicit a recovery response as compared to 1 μM of LA. One explanation for this is that LA at higher doses is fragmented into smaller peptide fragments that inhibit the reparative effects of LA. Indeed, analysis of the bathing solution containing LA revealed at least 4 fragments of LA were present, and application of a higher dose of F2 (10 μM) in particular inhibited the recovery response induced by LA. We also determined that F1 and F2 are formed by the brush border enzyme aminopeptidase M when AM was incubated with LA *in vitro*. Furthermore, we showed that AM is present in porcine mucosa, suggesting this is one of the enzymes responsible for the fragmentation of LA during *ex vivo* Ussing chamber experiments, which in turn leads to inhibition of recovery when LA is administered at higher doses (10 μM). Another fragment, F3, was also found from the *ex vivo* Ussing chamber experiments, suggesting another brush border enzyme, the proline specific endopeptidase carboxypeptidase P, may be involved in the formation of this fragment. Additionally, the formation of additional LA fragments beyond those presented in this manuscript is possible, and future studies involving the utilization of differential mobility spectrometry to improve limits of LC-MS detection may lead to detection and exploration of these other fragments [31]. The mechanisms by which these fragments inhibit recovery likely involve competition of LA binding sites on the epithelium, although these sites were not assessed in the present study.

The formation of inhibitory peptide fragments of LA during *ex vivo* recovery after intestinal ischemia presented both an interesting and impactful finding, as there is not only potential for these fragments to inhibit recovery of intestinal ischemia with higher doses of LA, but there is potential for these fragments to impact the efficacy of other injury models as well. Therefore, modification of LA to a D-amino acid analog was implemented as a method to potentially eliminate or slow fragmentation by aminopeptidase M or other brush border enzymes as the D-amino acid peptide would theoretically contain fewer specific substrates to be processed by brush-border enzymes as compared to the all-natural L-amino acid peptides of LA. In theory, D-amino acid larazotide, A6, would slow the buildup of inhibitory fragments. From the *ex vivo* stability data, it is evident that the D-amino acid A6 was significantly more stable for brush-border enzyme degradation than LA. From the *ex vivo* TER data presented in this study, it appears that modifying larazotide to a D-amino acid analog induces a recovery response similar to that of unaltered larazotide but at a 10-fold lower concentration of 0.1 μM. Consideration of when significant elevations in TER from each drug must be acknowledged, as the recovery response by A6 appeared to take effect earlier in the recovery period (T = 120 vs. T = 165 for LA).

Although chiral modification of LA to the D-amino acid analog A6 helped to reduce susceptibility to enzymatic digestion of this peptide drug, this chirally-modified form of LA can still be processed to fragments by brush-border enzymes such as AM and CP, resulting in the formation of three fragments of A6 (A6F1, A6F2, and A6F4) that were chemically synonymous to the formed LA fragments. In contrast to LA, the D-amino acid analog was primarily fragmented into A6F1, and this fragment was also found to be inhibitory. These findings further highlighted the complexity of peptide fragmentation, that may interfere with efficacy of LA or its D-amino acid analog. Nonetheless, a lower dose of A6 induced an earlier recovery response with noticeable differences in TER starting at 135-minutes of recovery as compared to LA. This suggests exploration of a D-amino acid analog of LA may have clinical utility.

## Conclusions

In conclusion, the present data demonstrate that LA elicits a recovery response after intestinal ischemic injury. This restorative capacity of LA on barrier function via TJ protein regulation after mucosal injury may be promising for other conditions that disrupt TJ proteins and result in paracellular barrier loss. However, fragmentation of larazotide presents some challenges because of apparent blockade of reparative effects of the parent compound. Future studies will assess modifications of the compound to address this issue, and the possibility of differing dosing regimens, such as multiple dosing rather than single dosing of tissue.

## Supporting information

**S1 Fig. Electrophysiological responses of ischemia-injured porcine jejunal mucosa to a multiple-dosing regimen of LA at various concentrations.** LA was applied to the apical bathing solution after the 30-minute equilibration period and every 45-minutes following the first dose. No statistical significance was detected between specific timepoint comparisons. Values are means ± SEM; n ≥ 3.
(TIF)

**S1 Raw images.**
(PDF)

## Author Contributions

**Conceptualization:** Zachary M. Slifer, Liliana Hernandez, Alexandra R. Carlson, Kristen M. Messenger, Jay Madan, B. Radha Krishnan, Sandeep Laumas, Anthony T. Blikslager.

**Data curation:** Zachary M. Slifer, Liliana Hernandez, Alexandra R. Carlson, Kristen M. Messenger, Anthony T. Blikslager.

**Formal analysis:** Zachary M. Slifer, Liliana Hernandez, Alexandra R. Carlson, Kristen M. Messenger, B. Radha Krishnan, Anthony T. Blikslager.

**Funding acquisition:** Jay Madan, Anthony T. Blikslager.

**Investigation:** Zachary M. Slifer, Liliana Hernandez, Tiffany A. Pridgen, Alexandra R. Carlson, Kristen M. Messenger, Jay Madan, B. Radha Krishnan, Anthony T. Blikslager.

**Methodology:** Zachary M. Slifer, Liliana Hernandez, Kristen M. Messenger, Jay Madan, B. Radha Krishnan, Anthony T. Blikslager.

**Project administration:** Kristen M. Messenger, Anthony T. Blikslager.

**Resources:** Tiffany A. Pridgen, Kristen M. Messenger, Jay Madan, B. Radha Krishnan, Anthony T. Blikslager.

**Software:** Tiffany A. Pridgen.

**Supervision:** Kristen M. Messenger, Anthony T. Blikslager.

**Validation:** Liliana Hernandez, Anthony T. Blikslager.

**Writing – original draft:** Zachary M. Slifer, Liliana Hernandez.

**Writing – review & editing:** Zachary M. Slifer, Liliana Hernandez, Alexandra R. Carlson, Kristen M. Messenger, Jay Madan, B. Radha Krishnan, Sandeep Laumas, Anthony T. Blikslager.

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
