## [Decision Letter · Decision Letter 0]

9 Feb 2021

PONE-D-21-00519

Larazotide acetate induces recovery of ischemia-injured porcine jejunum via repair of tight junctions

PLOS ONE

Dear Dr. Blikslager,

Thank you for submitting your manuscript to PLOS ONE. After careful consideration, we feel that it has merit but does not fully meet PLOS ONE’s publication criteria as it currently stands. Therefore, we invite you to submit a revised version of the manuscript that addresses the points raised during the review process.

Two reviewers have reviewed your manuscript.  While both find the manuscript is of interest for PLoS One readers, they have raised some concerns that you will need to address in a revised manuscript.

(1) Introduction.  The authors are required to add more backgrounds about LA including the important papers suggested by the reviewer 1.

(2) Evaluation of the LA’s effects on TJ by claudin-4.  Consider to evaluate the localization and amount of additional TJ transmembrane and/or plaque proteins, such as other claudins, occludin, JAMs, and ZO proteins.  Alternatively, the authors should at least justify the use of claudin-4 by providing the reason why claudin-4 can represent the effect(s) on TJs with appropriate citations.

(3) LPS flux values: The reviewer 2 raises a concern that the value of LPS flux is too small.  Please double-check the calculation.  Also, consider to use Papp, which is easily compared with the results from other papers.  Papp = (dQ/dt)/AxC0, where A = filter area, C0 = initial concentration in the donor compartment (See Van Itallie CM et al., J Cell Sci, 2007).

(4) As the reviewer 2 pointed out, the authors are recommended to improve the discussion of the results for the readers who are not familiar with the field.

We look forward to receiving your revised manuscript.

Kind regards,

Tomohito Higashi, Ph.D.

Academic Editor

PLOS ONE

Journal Requirements:

2.At this time, we request that you  please report additional details in your Methods section regarding animal care, as per our editorial guidelines:

(1) Please state the source and number of animals used in the study

(2) Please describe the post-operative care received by the animals, including the frequency of monitoring and the criteria used to assess animal health and well-being.

Thank you for your attention to these requests.

3. To comply with PLOS ONE submissions requirements, please provide the method of euthanasia in the Methods section of your manuscript.

4. Please note that PLOS does not permit references to “data not shown.” Authors should provide the relevant data within the manuscript, the Supporting Information files, or in a public repository. If the data are not a core part of the research study being presented, we ask that authors remove any references to these data.

5. Please ensure you have discussed any potential limitations of your study in the Discussion.

6. To comply with PLOS ONE submission guidelines, in your Methods section, please provide additional information regarding your statistical analyses. For more information on PLOS ONE's expectations for statistical reporting, please see https://journals.plos.org/plosone/s/submission-guidelines.#loc-statistical-reporting.

7. Please include your tables as part of your main manuscript and remove the individual files. Please note that supplementary tables (should remain/ be uploaded) as separate "supporting information" files.

8.PLOS ONE now requires that authors provide the original uncropped and unadjusted images underlying all blot or gel results reported in a submission’s figures or Supporting Information files. This policy and the journal’s other requirements for blot/gel reporting and figure preparation are described in detail at https://journals.plos.org/plosone/s/figures#loc-blot-and-gel-reporting-requirements and https://journals.plos.org/plosone/s/figures#loc-preparing-figures-from-image-files. When you submit your revised manuscript, please ensure that your figures adhere fully to these guidelines and provide the original underlying images for all blot or gel data reported in your submission. See the following link for instructions on providing the original image data: https://journals.plos.org/plosone/s/figures#loc-original-images-for-blots-and-gels.

9. Please amend the manuscript submission data (via Edit Submission) to include author Tiffany A. Pridgen.

10.Thank you for stating the following in the Competing Interests section:

"Anthony Blikslager and Zachary Slifer were funded, in part by Innovate Pharmaceuticals to perform this work. Anthony Blikslager consulted for Innovate Biopharmaceuticals Inc and 9 Meters Biopharma."

Reviewers' comments:

Reviewer's Responses to Questions

**Comments to the Author**

1. Is the manuscript technically sound, and do the data support the conclusions?

Reviewer #1: Yes

Reviewer #2: Yes

2. Has the statistical analysis been performed appropriately and rigorously? 

Reviewer #1: Yes

Reviewer #2: Yes

3. Have the authors made all data underlying the findings in their manuscript fully available?

Reviewer #1: Yes

Reviewer #2: Yes

4. Is the manuscript presented in an intelligible fashion and written in standard English?

Reviewer #1: Yes

Reviewer #2: Yes

5. Review Comments to the Author

Reviewer #1: In this submission to PLOS One, the authors present an experimental study to assess for the possibility of fragmentation of larazotide acetate (LA). The authors carry out western blot analysis of total protein isolated from uninjured and ischemia-injured porcine intestine, which showed aminopeptidase M enzyme presence in both tissue types, and mass spectrometry analysis of samples collected during ex vivo analysis confirmed formation of LA fragments. The authors conclude that LA stimulates repair of ischemic-injured epithelium at the

level of the tight junctions, at an optimal dose of 1 μM LA. The authors find that higher doses were less effective because of inhibition by LA fragments.

I find this manuscript to be of interest to readers of PLOS One, and I am supportive of publication with a minor note/revision. In particular, there has been complementary studies using mass spectrometry to understand analysis of fragments, which should be mentioned:

Journal of the American Society for Mass Spectrometry, 2014, 25, 1098-1113

Am J Gastroenterol. 2012, 107, 1554–1562

In particular, these prior studies have used mass spectrometry techniques to probe fragments as well as LA molecules in these studies. With this minor note/modification, I would be supportive of publication in PLOS One.

Reviewer #2: The authors should make efforts to describe what is known for the LA compound. As mentioned in the manuscript, LA is in Phase III Clinical trials.

Why the authors focus only on claudin-4? Tight junctions are complex with multiple types of claudins, occludin and junctional adhesion molecules. There is the matter of claudin shift, which indicates that a single claudin is not a good model.

pMLC is not well explained nor MLC or its phosphorylation.

LPS flux numbers are really small to be accurate measurements.

I can be more detailed in my review but essentially the discussion of the data was not clear for people that are outside the field nor for people new to the field.

6. PLOS authors have the option to publish the peer review history of their article (what does this mean?). If published, this will include your full peer review and any attached files.

Reviewer #1: No

Reviewer #2: **Yes: **dario mizrachi

---

## [Author Response · Author response to Decision Letter 0]

27 Mar 2021

Thank you to the editor and reviewers for the comments and critical feedback regarding this manuscript. All points from the editor and reviewers have been addressed in the attached "Response to Reviewers" document.

---

## [Editor Report · Decision Letter 1]

1 Apr 2021

Larazotide acetate induces recovery of ischemia-injured porcine jejunum via repair of tight junctions

PONE-D-21-00519R1

Dear Dr. Blikslager,

We’re pleased to inform you that your manuscript has been judged scientifically suitable for publication and will be formally accepted for publication once it meets all outstanding technical requirements.

Kind regards,

Tomohito Higashi, Ph.D.

Academic Editor

PLOS ONE

---

## [Editor Report · Acceptance letter]

12 Apr 2021

PONE-D-21-00519R1 

Larazotide acetate induces recovery of ischemia-injured porcine jejunum via repair of tight junctions 

Dear Dr. Blikslager:

I'm pleased to inform you that your manuscript has been deemed suitable for publication in PLOS ONE. Congratulations! Your manuscript is now with our production department. 

Kind regards, 

on behalf of

Dr. Tomohito Higashi 

Academic Editor

PLOS ONE